# 2015 versus 2021: Self-Reported Preparedness to Prescribe Antibiotics Prudently among Final Year Medical Students in Sweden

**DOI:** 10.3390/antibiotics13040303

**Published:** 2024-03-27

**Authors:** Jasmine Al-Nasir, Andrej Belančić, Dora Palčevski, Oliver J. Dyar

**Affiliations:** 1Department of Public Health and Caring Sciences, Uppsala University, 75236 Uppsala, Sweden; jasmineeal@outlook.com; 2Department of Clinical Pharmacology, Clinical Hospital Centre Rijeka, 51000 Rijeka, Croatia; andrej.belancic@uniri.hr; 3Department of Basic and Clinical Pharmacology with Toxicology, Faculty of Medicine, University of Rijeka, 51000 Rijeka, Croatia; 4Department of Internal Medicine, Clinical Hospital Centre Rijeka, 51000 Rijeka, Croatia; dora.palcevski@gmail.com

**Keywords:** antimicrobials, antimicrobial stewardship, medical education

## Abstract

Cross-sectional surveys have found variations in how prepared medical students feel to prescribe antibiotics responsibly, but insights are lacking on the stability of these outcomes. In a 2015 survey, final-year Swedish medical students reported very high preparedness levels across a comprehensive range of relevant curriculum topics. We repeated this survey in 2021 to assess the stability of previous findings and to capture the potential impacts of the COVID-19 pandemic. Final-year students in 2015 and 2021 at all seven Swedish medical schools were eligible to participate in an online survey covering curricula topics, teaching methods and COVID-19 impacts (2021). Eligible students received email invitations and reminders from local coordinators. Students from six of seven medical schools participated in both surveys, with response rates of 24.1% (309/1281) in 2021 and 21.3% (239/1124) in 2015. The average global preparedness was 77.0% and 83.2%, respectively (*p* < 0.001), with lower preparedness levels in 24/27 curriculum topics in 2021. Students at certain universities reported COVID-19 impacts on antibiotic prescribing education (format, duration and perceived quality). Self-reported preparedness levels have fallen slightly but remain high compared with 2015 levels in other European countries. Students consistently reported lower preparedness in specific topics; improvement efforts should consider focusing on these areas, particularly in the context of the ongoing implementation of programmes leading to a full licence upon graduation.

## 1. Introduction

Prescribing drugs is a core activity in the daily work of most medical doctors, yet a growing number of recent studies indicate that many, if not most, medical graduates do not feel sufficiently prepared for their upcoming prescribing responsibilities [1,2,3,4,5,6]. Since junior doctors are responsible for a considerable share of drug prescriptions, this is a serious concern for medical educators [2,3]. The importance of improving undergraduate education of medical students in responsible antibiotic prescribing is a key effort in addressing the growing mortality burden attributed to antimicrobial resistance (AMR), particularly in the context of studies indicating persistent antibiotic overuse and misuse in healthcare settings [7,8,9,10]. Education to improve responsible antibiotic use is thus considered a core component of antibiotic stewardship, which has been defined as a *coherent set of actions that promote using antimicrobials responsibly* [11]. Such actions include a combination of both restrictive actions (e.g., formulary restrictions and pre-approval requirements) and enablement actions (education, clinical guidelines, audit and feedback) [12].

Strong education in responsible antibiotic use and the broader factors leading to antibiotic misuse and overuse can, at the undergraduate level, lay an important foundation for students to contribute to future antibiotic stewardship efforts as well as to understand their purpose. Medical school is a crucial time during which knowledge, attitudes and behaviours of future prescribers are still being shaped, both by the explicit curriculum and through the hidden curriculum of prescribing practices they are exposed to on clinical placements and rotations [13]. It is, therefore, natural that the World Health Organization (in the Global Action Plan on Antimicrobial Resistance) recommends that educational efforts should target undergraduates, not just postgraduate education, stating that “making antimicrobial resistance a core component of professional education, training, certification, continuing education and development in the health and veterinary sectors and agricultural practice will help to ensure proper understanding and awareness among professionals.” [14]. Developing an understanding of the current state of undergraduate education in responsible antibiotic use is an essential component of efforts to improve education.

### 1.1. Antimicrobial Stewardship in Sweden

Sweden has a strong history of raising public awareness of antibiotic resistance and implementing efforts to reduce the use and misuse of antibiotics. Antibiotic stewardship initiatives, coordinated initially by Strama, a nationwide working model to address antibiotic resistance in health care, have existed for many years at national and regional levels and have placed a strong emphasis on patterns of antibiotic prescribing in primary care [15]. A Swedish national action plan was first published in 2000 and has been revised several times, with professional education mentioned in many versions. The 2016 plan [16], following on from the Global Action Plan, stated that the “government expects relevant staff to have knowledge about antibiotic resistance, the spread of infectious diseases and the importance of a high degree of compliance with basic hygiene routines and other infection prevention measures, as well as knowledge about the seriousness and complexity of the issue from a global perspective” and further that “the government expects antibiotic resistance, infection prevention and control/hygiene to be included in relevant education and training programmes”.

A consequence of longstanding efforts to minimise antibiotic misuse and overuse in Sweden is that surveillance studies have regularly shown relatively low levels of antibiotic consumption, high proportions of narrow-spectrum antibiotic use and low resistance rates among key pathogens [9,10,15].

### 1.2. Aim

Previous cross-sectional surveys suggest that medical students in Europe and beyond do not feel sufficiently well prepared to prescribe antibiotics responsibly. A key limitation of previous studies is that they have typically captured data at a single time point [4,17,18,19], meaning that insights are lacking on the longer-term *stability* of outcomes from such studies; evidence of stable preparedness levels in the absence of widespread changes in teaching would further validate the ability to draw conclusions from studies conducted at a single time point. In addition, many previous investigations have only assessed a narrow range of curriculum topics relevant to responsible antibiotic use. In 2015, we conducted a large-scale survey study of final-year medical students at all European medical schools [4]. Compared with other respondents, Swedish medical students reported very high preparedness levels across a comprehensive range of curriculum topics in responsible antibiotic prescribing and low levels of perceived need for further education. The main aim of the present study was to build on the previous literature by assessing the stability of these previous findings, both for overall preparedness levels and preparedness in individual topics, by repeating the same 2015 survey with final-year medical students in Sweden in 2021. In addition, we aimed to capture the potential impacts of the COVID-19 pandemic on education and preparedness within responsible antibiotic use. Our primary research questions were: Are the previously observed high overall preparedness levels among final-year Swedish students stable (i.e., similar in 2021 to 2015)? Have there been widespread improvements or declines in preparedness levels for individual curriculum topics? Do final-year students believe that the COVID-19 pandemic has affected the form, quality and duration of education on antibiotic prescribing at medical school?

## 2. Results

### 2.1. Participation

A total of 239 and 309 eligible responses were received in 2015 and 2021, respectively, representing response rates of 21.3% (239/1124) and 24.1% (309/1281). Response rates per individual medical school are presented in Appendix A. No significant difference was found in gender distribution between the two years (57.3% vs. 63.7% female; *p* = 0.128), but the average age of participating students was lower in 2021 than in 2015 (52.4% vs. 37.0% ≤25 years old, *p* < 0.001).

### 2.2. Global Preparedness Scores

The country-level global preparedness score (GPS), representing the average percentage of topics for which students felt at least sufficiently prepared, was slightly higher in 2015 (83.2% vs. 77.0%, *p* < 0.001). The medical school GPS ranged from 80.5–85.9% in 2015 and 69.3–82.5% in 2021, as presented in Figure 1, with lower GPS scores at every medical school in 2021. Statistically significant differences (Appendix A) were observed for Örebro (11.5% difference), Umeå (13% difference) and Karolinska (3.5% difference).

### 2.3. Topic Preparedness Scores

Students reported lower preparedness levels in 2021 than in 2015 for 24 of the 27 curriculum topics included in the study instrument, with 10 of these differences being statistically significant (Table 1). The only questions for which students reported being more prepared in 2021 (although non-significant) were: ‘To recognise the clinical signs of infection’ (+0.2% difference), ‘To interpret biochemical markers of inflammation’ (+0.9% difference) and ‘To work within the multi-disciplinary team in managing antibiotic use in hospitals’ (+2.2% difference). The absolute percentage changes in topic preparedness scores between studied years per medical school are shown in Appendix A. The curriculum topic for which students felt the least prepared was ‘To prescribe using principles of surgical antibiotic prophylaxis’ both in 2015 and 2021 (53.8% vs. 44.0%, *p* = 0.020), whilst they felt the best prepared ‘To recognise the clinical signs of infection’ (99.2% vs. 99.4%, *p* = 0.700). The topics with the highest and lowest preparedness and greatest variations between medical schools for 2021 are shown in Table 2. Table 3 shows the two topics with the lowest preparedness scores at each medical school in 2021.

### 2.4. Expressed Need for Further Education

The percentages of students at each medical school who felt they needed more education on prudent antibiotic use are presented in Table 4. Students expressed greater needs for more education in 2021 than in 2015 on average (36% vs. 21%, *p* = 0.003), with the same trend observed at every medical school. The greatest between-survey difference was at Örebro (34% difference, *p* = 0.023), and the least was for Uppsala (3% difference, *p* = 0.706).

### 2.5. Impacts of COVID-19 on Education

Respondents in the 2021 survey expressed that COVID-19 had negatively affected their teaching of prudent antibiotic use in terms of format (32.4%, 94/290), duration (12.0%, 35/291) and quality (16.5%, 47/284). Sixteen per cent of respondents (45/286) felt that the impacts of the COVID-19 pandemic on their education at medical school had worsened how prepared they felt to prescribe antibiotics prudently. Widespread variations were seen between medical schools.

Fourteen per cent (39/273) of students reported having received teaching on prescribing antibiotics for patients with COVID-19, either primarily by clinical supervisors (51%, 20/39) or at lectures, seminars or case-based discussions (49%, 19/39).

### 2.6. Perceived Usefulness of Teaching Methods

Table 5 presents the perceived usefulness of different teaching methods among 2015 and 2021 respondents, including variations between medical schools. The most useful teaching method was considered to be ‘small group teaching’ both in 2015 and 2021, and the least useful teaching methods were considered to be ‘microbiology clinical placement’ and ‘active learning assignments’. In addition, the teaching methods considered to be least available/used were ‘microbiology clinical placement’ and ‘pear or near-peer teaching’.

## 3. Discussion

We conducted a repeated cross-sectional questionnaire survey of final-year medical students in Sweden to assess changes in preparedness to prescribe antibiotics responsibly between 2015 and 2021. Our aim was to assess whether the previously high levels of preparedness were stable in the context of factors that may have influenced preparedness levels, such as rising global awareness of AMR with the Global Action Plan [14], a new Swedish National Action Plan in 2016 [16] and the impacts of the COVID-19 pandemic [20].

### 3.1. Overall Preparedness Levels and Needs for Further Education

Our overall results show that medical students in Sweden had significantly lower overall self-reported preparedness levels in 2021 compared to 2015, with trends towards lower preparedness levels seen in all medical schools and across almost all curriculum topics. In keeping with lower reported preparedness levels, we also found increased proportions of students reporting a need for more education on prudent antibiotic use at all medical schools in 2021.

It is thus clear that there has not been any large-scale improvement in preparedness despite the growing awareness of antimicrobial resistance and the development and implementation of both the Global Action Plan [14] and the National Action Plan [16]. Although both of these plans stressed the importance of undergraduate professional education, to our knowledge, there have been no systematic or top-down efforts to improve education in responsible antibiotic use in medical education since 2015. One potential explanation for this lack of effort is that data on antibiotic prescribing suggest that doctors in Sweden already have very low levels of antibiotic misuse and overuse, in international terms, and current approaches in undergraduate education may have been considered sufficient. Our study methodology does not allow us to explain the observed lowering of preparedness levels and raising of perceived needs for more education from 2015 to 2021. One possibility is that more widespread awareness of the perceived threat of AMR, in part in response to global and national action plans, may have negatively influenced students’ perceptions of their own preparedness to prescribe antibiotics responsibly.

Many students reported impacts on the form, quality and duration of education on antibiotic prescribing caused by the COVID-19 pandemic. Medical schools in Sweden, as in the rest of the world, had to shift from clinical placements to predominantly virtual education during the pandemic, with potential consequences on the quality of teaching and learning [20,21,22]. Even if the majority of students in this study had already had their clinical rotations in infectious diseases before the pandemic, a reduction in exposure to daily antibiotic prescribing practices on other clinical placements could be a contributing factor to the observed lower perceived preparedness levels. This hypothesis is supported by a previous study in which medical students from Gothenburg reported that clinical rotations were the element considered to be most valuable for learning, as well as the chance to practice prescribing medications under supervision in clinical wards [18]. Interestingly, one of the curriculum topics with the greatest reduction in preparedness levels between 2021 and 2015 was *differentiating between bacterial and viral respiratory tract infections*. This result could be a result of a greater awareness of the complexity of the task of distinguishing between viral and bacterial infections gained during the pandemic. It is possible that the previous very high levels of preparedness on this topic (93%) actually represented a degree of over-confidence among students, so a lowering in perceived preparedness levels in this topic is not necessarily negative. This is in line with studies showing that during the early stages of the COVID-19 pandemic, patients often received empirical antibiotics even though they did not have a bacterial infection, mainly because the symptoms of bacterial infections are similar to viral infections [23].

### 3.2. Consistency in Relative Topic Preparedness Scores and Teaching Methods

The ranking of topic preparedness levels, i.e., from high to low preparedness levels, was quite consistent between medical schools in both 2015 and 2021 and is largely in keeping with the overall ranking of curriculum topics observed across all European medical schools in 2015 [4]. A similar study using an almost identical questionnaire was recently conducted with medical students in Croatia and found that results on preparedness on certain topics were largely consistent between 2015 and 2019 [5]. This consistency, despite variations in medical school curricula, socioeconomic features and cultures, suggests that certain topics are either inherently more difficult to feel prepared for or are consistently less well taught than others.

Similarly, we identified no large changes between 2015 and 2021 in terms of the teaching methods used or their perceived usefulness in the context of learning about responsible antibiotic use. Small group teaching, discussion of clinical cases and infectious diseases clinical placements were considered to be most useful, a pattern seen across all European medical schools in 2015 [4] and also in the repeated cross-sectional study among students in Croatia in 2019 [5]. A number of medical programmes outside Sweden are starting to use externally produced virtual courses, such as the WHO-online antibiotic stewardship course, and it will be interesting to assess the longer-term impact of these teaching methods both in terms of resource-saving potential and student preparedness [24].

### 3.3. Variations between and within Medical Schools

Our primary aim was to assess the stability of results at a national level in Sweden by repeating our survey with the same instrument and at the same time point in the medical programs. Although we did detect a number of statistically significant differences at individual medical schools between the two survey years, our study was likely underpowered to detect many potential variations (for instance, at the level of preparedness in individual topics). We also purposely avoided comparing preparedness results between medical schools, as our study was not designed to facilitate such comparisons. The level of perceived preparedness may be affected by *recency* in exposure to core teaching in infection diseases and antibiotics. The placement of this course varied from Term 5 (Gothenburg) to Term 9 (Uppsala), which may have influenced the feeling of preparedness and our results. This is challenging to assess, however, given the variations in exposure that individual students will have both through the hidden curriculum (i.e., exposure to daily clinical practices on other clinical rotations) as well as through case-based discussions as part of other clinical rotations, which may involve consideration of antibiotic therapy. It is also important to note that one medical school, Örebro, is a relatively new medical school. The first students graduated soon after the 2015 survey, so particular caution needs to be taken when comparing variations between survey years at Örebro with changes observed between survey years at other medical schools.

Finally, it is important to note that a comprehensive review of changes at individual medical schools in curricula, teaching sessions and examinations between 2015 and 2021 was beyond the scope of the present study. Such a review is to be encouraged at participating medical schools to allow a context-specific interpretation of the results presented here, particularly regarding changes in preparedness in individual curricula topics.

### 3.4. Methodological Considerations

Key strengths of our study include the use of an identical study design and comprehensive questionnaire, in order to allow comparison between two cohorts whilst minimizing the introduction of additional sources of bias. Since our primary aim was to assess the *stability* of self-reported preparedness levels between 2015 and 2021, we did not choose to include additional questions covering topic areas that may now appear in medical school curricula within antimicrobial stewardship (e.g., the concept of One Health and the WHO AWaRe classification of antibiotics). Our response rates were consistent between years but, as with many other studies in the same subject area [4,6,25], somewhat low. We have no data available on reasons for non-participation. We consider that even if the response rates were low, our results are likely representative of all medical schools, as no correlation was found between response rate and preparedness levels. Furthermore, the similarity of response rates between survey years provides some indication that any effects of selection biases (e.g., risks of over-representation of students with strongest opinions) were similar during both survey periods. We have previously noted [4,18] that an important consideration when interpreting our findings is that the questionnaire was designed to elicit students’ *self-reported* preparedness levels within each curriculum topic, as opposed to objectively assessing individual students’ *actual* preparedness levels. This, therefore, limits the interpretations that can be directly made from our results; additional studies are required to understand how more objective assessments (e.g., with case vignettes) and self-reported preparedness levels are related and whether the strength of the relationship varies between different topics. Nonetheless, we contend that gaining student perspectives on teaching is a vital component of quality improvement efforts and is fully in keeping with recommended student-centered practices in higher education [26]. Finally, it is also important to note that observational questionnaire studies such as ours, even if conducted anonymously, may be influenced by a social-desirability bias among respondents; however, we have no reason to suspect that this potential bias would disproportionately affect data collection from a particular study period, so it should not influence our ability to assess *stability* in self-reported preparedness levels between survey years.

## 4. Materials and Methods

### 4.1. Study Design

To assess medical students’ self-reported preparedness for responsible use of antibiotics, the European Society of Clinical Microbiology and Infectious Diseases (ESCMID) Study Group for Antimicrobial Stewardship (ESGAP) conducted a cross-sectional, multicentre, web-based survey for final year medical students at medical schools in 29 European countries in 2015 [4]. The same study design was repeated in 2021 in Sweden at all seven medical schools with a modified questionnaire tool.

### 4.2. Medical Schools in Sweden

There are seven medical schools in Sweden, located in Stockholm, Gothenburg, Uppsala, Lund, Linköping, Umeå and Örebro. A set of seventeen national “examination goals” exist and every medical school is legally required to ensure that their own curriculum ensures that graduating students reach these learning outcomes. Importantly, these learning outcomes consist of high-level descriptions of the *Knowledge and understanding*, *Skills and abilities*, and *Judgment and approach* that students should develop. This means that within a specific subject area, such as responsible antibiotic use, there can be variations between medical schools in terms of the curricula content, teaching methods used and examinations. For instance, in Linköping and Örebro, *problem-based learning* is the main teaching method throughout the medical programme, whereas *flipped classroom case-based learning* is the most used teaching activity at Lund. At the time of the survey, all medical programmes were required to be 5.5 years (11 terms) in duration. The structure, however, varies between universities, including the time points at which specific subjects and clinical placements are included. For instance, the infectious diseases course takes place during different terms: Gothenburg (during the 5th term), Karolinska Institute (5th and 6th terms), Lund (7th term), Umeå (8th term), Uppsala (9th term) and Örebro (7th and 8th terms).

### 4.3. Survey Development

The survey instrument was developed in 2015 by a group of six international experts on antimicrobial stewardship through an informal consensus process based both on a previous study of undergraduate curricula coverage of principles of prudent antibiotic use in European medical schools [13] and a review of relevant previous studies among medical students. Its complete design, development and piloting (with students from two countries) have been extensively described elsewhere [4]. The 47-item study instrument included questions on demographics, self-reported preparedness in 27 curricular topics on prudent antibiotic use (based on a 7-point Likert-type scale principle), opinions of the usefulness of a selection of commonly used teaching methods in undergraduate medical education, and perceived need for further education on this topic. The survey instrument was revised before use in 2021 through the development and addition of nine questions concerning the impact of the COVID-19 pandemic on education in responsible antibiotic use, included at the end of the 2015 version of the questionnaire. These questions were developed by the first and last author through a literature review in 2021 to identify other studies that had investigated the impacts of COVID-19 on medical student teaching and learning, and the questions were constructed to follow a similar format to the other questions in the original questionnaire. The additional questions were tested for face validity but were not otherwise subjected to pilot testing, given the low complexity level of the questions.

### 4.4. Survey Distribution and Participants

We followed the same steps for recruitment of participants and survey distribution for the 2021 survey as in the study protocol from 2015 [4]: the “program committee” at each medical school was contacted during early autumn 2021 to identify a local coordinator with responsibility for sending invitations to all eligible medical students, i.e., students in their final term or second last term of medical school at a Swedish university during autumn 2021. Template invitations were drafted by the central study coordinator and forwarded to medical school coordinators for distribution via email to students. The medical school coordinators were also asked to provide the exact number of eligible students per term, in order to be able to calculate response rates. The initial invitation with a survey link was sent by each medical school coordinator by email to all eligible medical students, and this was followed by two email reminders after 2–3 and 4–6 weeks. The self-administered survey was accessible via SurveyMonkey^®^ (SurveyMonkey Inc., San Mateo, CA, USA) during spring 2015 (for approximately five months) and autumn 2021 (for approximately three months). Informed consent was obtained from all subjects involved in the study. Participation was voluntary and anonymous.

### 4.5. Statistical Analyses

Data was exported from the SurveyMonkey^®^ service, and the analyses were performed using Microsoft Excel 2019^®^ (Microsoft Office). Responses were excluded from subsequent analyses if (i) the response rate at the medical school was lower than the pre-specified cut-off of 10% in 2015 and 2021 (leading to the removal of Linköping medical school) or (ii) a respondent did not answer all of the preparedness questions. Responses to questions about preparedness for each of the 27 topics covered in the curriculum were divided into two categories: (4–7) “*at least sufficiently prepared*” and (1–3) “*insufficiently prepared*”.

“Topic preparedness scores” and “global preparedness scores” (GPS) were calculated. The topic preparedness score represents the percentage of medical school students at an individual medical school who felt at least sufficiently prepared for an individual topic. The global preparedness score represents the average of the 27 topic preparedness scores at each medical school. National-level topic preparedness scores and global preparedness scores were also calculated, with each medical school being weighted equally.

Data were presented as measures of central tendency and measures of spread as well as absolute and relative frequencies. The Kolmogorov–Smirnov (KS) test was used to assess the normality of distribution. Comparisons between the availability of teaching methods according to the year of graduation (2015 vs. 2021), the perceived needs for further education and topic/global preparedness scores were conducted using a chi-square test. The COVID-19 and teaching methods questions were presented via descriptive statistical analyses. Statistical significance was set at *p* < 0.05.

## 5. Conclusions

Self-reported preparedness levels in rational antibiotic prescribing fell slightly between 2015 and 2021, but students in Sweden still feel well-prepared compared with students in other European countries, at least according to 2015 levels. Students consistently reported low levels of preparedness in specific curricula topics, and future improvement efforts should consider focusing on these areas. COVID-19 likely had at least some negative impacts on teaching about responsible antibiotic prescribing, and the long-term consequences of these effects deserve follow-up.

More broadly, as all medical schools in Sweden now continue efforts to introduce a six-year (twelve-term) medical programme, we hope that our results can help stimulate and facilitate reflection on the content and form of teaching about responsible antibiotic use. From spring 2027, students will graduate with a full medical licence and, thus, greater expectations for competence and independence.

## Figures and Tables

**Figure 1 antibiotics-13-00303-f001:**
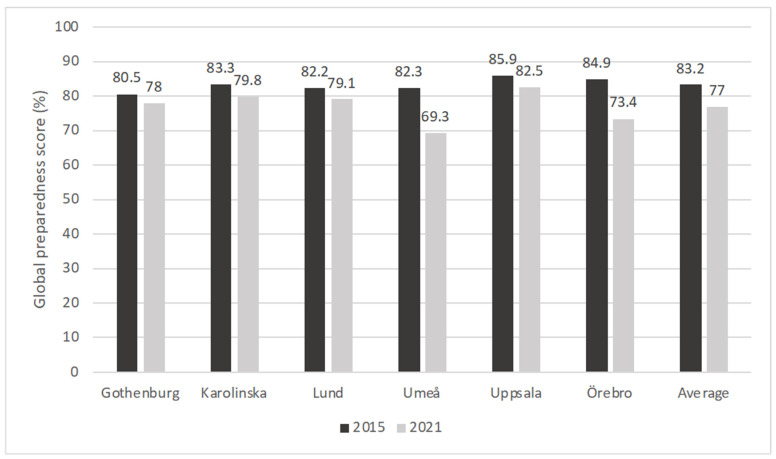
Global preparedness scores for each medical school in 2015 and 2021.

**Table 1 antibiotics-13-00303-t001:** Average preparedness score in 2021 compared to 2015 for each topic. The difference is shown in the table and the range in difference between medical schools.

Topic	2015 (*n* = 239)		2021 (*n* = 309)				
	Sufficiently Prepared		Sufficiently Prepared		Difference ^1^	*p*	Trend
	%	Range	%	Range	(Range)		
To recognise the clinical signs of infection	99.2	97–100	99.4	97–100	0.2 (−0.3–0.8)	0.7	↗
To assess the clinical severity of infection (e.g., using criteria such as the septic shock criteria)	95.0	92–100	92.2	89–98	−2.8 (−7.5–5.9)	0.2	↘
To use point-of-care tests (e.g., urine dipstick, rapid diagnostic tests for streptococcal pharyngitis)	89.9	80–94	87.3	72–98	−2.6 (−14.8–4.4)	0.3	↘
To interpret biochemical markers of inflammation (e.g., CRP)	97.1	92–100	98.0	94–100	0.9 (−1.7–4.3)	0.5	↗
To decide when it is important to take microbiological samples before starting antibiotic therapy	95.4	92–97	94.1	85–98	−1.3 (−6.9–1.0)	0.5	↘
To interpret basic microbiological investigations (e.g., blood cultures, antibiotic susceptibility reporting)	93.7	86–97	92.8	85–100	−0.9 (−6.8–6.0)	0.7	↘
To identify clinical situations when not to prescribe an antibiotic	93.7	92–97	84.4	64–90	−9.3 (−30.2–−2.6)	**<0.001**	↘
To differentiate between bacterial colonisation and infection (e.g., asymptomatic bacteriuria)	91.2	89–97	87.3	75–95	−3.9 (−19.1–2.1)	0.1	↘
To differentiate between bacterial and viral upper respiratory tract infections	93.3	86–100	81.9	75–88	−11.4 (−25.0–−2.6)	**<0.001**	↘
To select initial empirical therapy based on the most likely pathogen(s) and antibiotic resistance patterns without using guidelines	71.4	64–82	61.8	49–78	−9.6 (−26.6–2.6)	**0.02**	↘
To decide the urgency of antibiotic administration in different situations (e.g., <1 h for severe sepsis, non-urgent for chronic bone infections)	87.9	76–94	82.8	77–90	−5.1 (−17.7–9.4)	0.1	↘
To prescribe antibiotic therapy according to national/local guidelines	92.4	82–97	87.3	75–94	−5.1 (−11.6–3.6)	0.05	↘
To assess antibiotic allergies (e.g., differentiating between anaphylaxis and hypersensitivity)	76.1	66–94	72.3	60–91	−3.8 (−18.5–25.5)	0.3	↘
To identify indications for combination antibiotic therapy	61.4	48–70	48.3	32–62	−13.1 (−25.8–−5.4)	**0.002**	↘
To decide the shortest possible adequate duration of antibiotic therapy for a specific infection	62.2	46–76	48.2	36–57	−14.0 (−31.7–0.4)	**0.001**	↘
To prescribe using principles of surgical antibiotic prophylaxis	53.8	35–62	44.0	32–51	−9.8 (−25.8–−5.4)	**0.02**	↘
To review the need to continue or change antibiotic therapy after 48–72 h, based on clinical evolution and laboratory results	81.2	71–94	75.4	62–89	−5.8 (−25.6–11.9)	0.1	↘
To assess clinical outcomes and possible reasons for the failure of antibiotic treatment	83.1	76–88	77.5	66–87	−5.6 (−12.3–−1.2)	0.1	↘
To decide when to switch from intravenous (IV) to oral antibiotic therapy	75.1	68–82	65.2	51–75	−9.9 (−22.0–−4.2)	**0.01**	↘
To measure/audit antibiotic use in a clinical setting and to interpret the results of such studies	64.4	44–57	50.2	21–43	−14.2 (−30.6–−7.8)	**<0.001**	↘
To work within the multi-disciplinary team in managing antibiotic use in hospitals	69.3	54–70	71.5	42–71	2.2 (−17.2–5.4)	0.6	↗
To discuss antibiotic use with patients who are asking for antibiotics when I feel they are not necessary	95.3	88–97	93.4	81–98	−1.9 (−7.7–3.9)	0.3	↘
To communicate with senior doctors in situations where I feel antibiotics are not necessary, but I feel I am being inappropriately pressured into prescribing antibiotics by senior doctors	60.9	46–71	47.3	22–47	−13.6 (−48.4–−0.4)	**0.002**	↘
To use knowledge of the common mechanisms of antibiotic resistance in pathogens	84.0	78–100	78.8	62–90	−5.2 (−30.9–5.9)	0.1	↘
To use knowledge of the epidemiology of bacterial resistance, including local/regional variations	75.5	69–82	62.1	38–78	−13.4 (−40.6–−3.5)	**<0.001**	↘
To practise effective Infection control and hygiene (to prevent the spread of bacteria)	97.9	95–100	95.5	89–98	−2.4 (−6.6–−0.3)	0.1	↘
To use knowledge of the negative consequences of antibiotic use (bacterial resistance, toxic/adverse effects, cost, *Clostridium difficile* infections)	97.9	94–100	95.8	89–98	−2.1 (−8.6–3.1)	0.2	↘

^1^ A negative result indicates a lower score in 2021 than in 2015.

**Table 2 antibiotics-13-00303-t002:** Curriculum topics with the highest and lowest preparedness scores and greatest variation between medical schools in 2021.

Highest Preparedness	Lowest Preparedness	Greatest Variation between Medical Schools
To recognize the clinical signs of infection (99.4%)	To prescribe using principles of surgical antibiotic prophylaxis (44.0%)	To use knowledge of the negative consequences of antibiotic use (bacterial resistance, toxic/adverse effects, cost, *Clostridium difficile* infections) (37.7–78.3%)
To interpret biochemical markers of inflammation (e.g., CRP) (98.0%)	To communicate with senior doctors in situations where I feel antibiotics are not necessary but I feel I am being inappropriately pressured into prescribing antibiotics by senior doctors (47.3%)	To use knowledge of the common mechanisms of antibiotic resistance in pathogens (60.4–91.5%)
To use knowledge of the negative consequences of antibiotic use (bacterial resistance, toxic/adverse effects, cost, *Clostridium difficile* infections) (95.8%)	To decide the shortest possible adequate duration of antibiotic therapy for a specific infection (48.2%)	To interpret basic microbiological investigations (e.g., blood cultures, antibiotic susceptibility reporting) (41.7–71.4%)
To practise effective Infection control and hygiene (to prevent the spread of bacteria) (95.5%)	To identify indications for combination antibiotic therapy (48.3%)	To differentiate between bacterial and viral upper respiratory tract infections (32.1–61.7%)
To decide when it is important to take microbiological samples before starting antibiotic therapy (94.1%)	To measure/audit antibiotic use in a clinical setting and to interpret the results of such studies (50.2%)	To assess antibiotic allergies (e.g., differentiating between anaphylaxis and hypersensitivity) (49.1–78.3%)

**Table 3 antibiotics-13-00303-t003:** Curriculum topics with the lowest preparedness score per medical school in 2021.

Gothenburg	Karolinska	Lund	Umeå	Uppsala	Örebro
To prescribe using principles of surgical antibiotic prophylaxis (36.2%)	To communicate with senior doctors in situations where I feel antibiotics are not necessary but I feel I am being inappropriately pressured into prescribing antibiotics by senior doctors (34.7%)	To measure/audit antibiotic use in a clinical setting and to interpret the results of such studies (40.6%)	To measure/audit antibiotic use in a clinical setting and to interpret the results of such studies (20.8%)	To prescribe using principles of surgical antibiotic prophylaxis (40.0%)	To communicate with senior doctors in situations where I feel antibiotics are not necessary but I feel I am being inappropriately pressured into prescribing antibiotics by senior doctors (22.2%)
To measure/audit antibiotic use in a clinical setting and to interpret the results of such studies (36.2%)	To measure/audit antibiotic use in a clinical setting and to interpret the results of such studies (38.8%)	To communicate with senior doctors in situations where I feel antibiotics are not necessary but I feel I am being inappropriately pressured into prescribing antibiotics by senior doctors (45.3%)	To prescribe using principles of surgical antibiotic prophylaxis (32.1%)	To measure/audit antibiotic use in a clinical setting and to interpret the results of such studies (43.3%)	To measure/audit antibiotic use in a clinical setting and to interpret the results of such studies (27.8%)

**Table 4 antibiotics-13-00303-t004:** Students’ expressed needs for more education on responsible antibiotic use in 2015 and 2021.

Percentage of Students Who Feel They Need More Education	2015	2021	*p*
Gothenburg University	18%	36%	0.05
Karolinska Institutet	25%	40%	0.09
Lund University	19%	32%	0.18
Umeå University	32%	37%	0.63
Uppsala University	13%	16%	0.7
Örebro University	18%	52%	**0.02**
Average	21%	36%	**0.003**

**Table 5 antibiotics-13-00303-t005:** Students’ perceived usefulness of different teaching methods for responsible antibiotic use.

Teaching Method	Useful or Very Useful				Not Useful		Neutral		I Am Unsure		I Do Not Understand the Question		Teaching Method Was Not Used ^1^	
	2015		2021		2015	2021	2015	2021	2015	2021	2015	2021	2015	2021
	%	Range	%	Range	%	%	%	%	%	%	%	%	%	%
Lectures (with >15 people)	88.1	77.1–97.0	77.7	57.8–86.7	0.4	0.0	10.2	19.1	1.3	3.2	0.0	0.0	0.4	2.1
Small group teaching (with <15 people)	95.0	70.6–93.9	91.0	55.6–90.6	0.9	0.7	2.7	3.7	1.4	4.5	0.0	0.0	5.5	7.9
Discussions of clinical cases and vignettes	92.2	82.4–92.5	90.9	66.7–92.2	0.4	0.3	4.8	5.9	2.2	2.1	0.4	0.7	1.3	1.0
Active learning assignments	51.9	35.1–52.0	54.1	25.0–47.2	7.4	6.4	35.4	32.7	5.3	6.8	0.0	0.0	18.9	23.9
Infectious diseases clinical placement	93.6	88.6–95.5	82.6	58.3–85.9	0.4	3.2	5.1	12.4	0.8	1.1	0.0	0.7	0.0	2.8
Microbiology clinical placement	47.1	8.6–52.9	38.8	8.3–29.8	10.1	6.2	30.4	39.5	11.6	14.7	0.7	0.8	41.5	55.2
Peer or near-peer teaching	68.5	27.3–48.6	67.5	22.2–42.2	1.5	1.3	17.7	20.5	10.8	9.9	1.5	0.7	44.7	47.8

^1^ Participants who responded that a teaching method was not used were not included in denominators for assessing the usefulness of a teaching method (i.e., all the other results in the table).

## Data Availability

The data presented in this study are available on request from the corresponding author.

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
