# Peer review of "2015 versus 2021: Self-Reported Preparedness to Prescribe Antibiotics Prudently among Final Year Medical Students in Sweden"

_antibiotics, 2024, doi:10.3390/antibiotics13040303_

Round 1
Reviewer 1 Report
Comments and Suggestions for Authors
Considering the antimicrobial resistance emergence as an even more current issue to deal with, this manuscript presents some interesting aspects on self-reported antimicrobial prescription preparedness among Swedish medical students. The found results can be of public concern, but the following minor suggestion can improve your study:
- Introduce the concept of Antimicrobial Stewardship in the Introduction section;
- Explain better the consistence of Global Action Plan and National Action Plan in Discussion section;
- Add more details about the followed methods for the literature review process and relative references in "Survey Development" in Materials and Methods section
Author Response
Thank you for your helpful comments, we have now revised the manuscript in line with these.
Reviewer one:
Considering the antimicrobial resistance emergence as an even more current issue to deal with, this manuscript presents some interesting aspects on self-reported antimicrobial prescription preparedness among Swedish medical students. The found results can be of public concern, but the following minor suggestion can improve your study:
- Introduce the concept of Antimicrobial Stewardship in the Introduction section;
Response: Thank you for this suggestion, we have added a couple of sentences to the introduction to introduce the concept of stewardship more clearly:
Education to improve responsible antibiotic use is thus considered a core component of antibiotic stewardship, which has been defined as a coherent set of actions which promote using antimicorbials responsibly [11]. Such actions include a combination of both restrictive actions (e.g. formulary restrictions, pre-approval requirements) and enablement actions (education, clinical guidelines, audit and feedback) [12]. Strong education in responsible antibiotic use, and the broader factors leading to antibiotic misuse and overuse, can at the undergraduate level lay an important foundation for students to contribute to future antibiotic stewardship efforts – as well as to understand their purpose. Medical school is a crucial time during which knowledge, attitudes, and behaviours of future prescribers are still being shaped, both by the explicit curriculum and through the hidden curriculum of prescribing practices they are exposed to on clinical placements and rotations [13].
- Explain better the consistence of Global Action Plan and National Action Plan in Discussion section;
Response: We have now added additional text both in the introduction and discussion concerning the relevance of these action plans.
Introduction:
It is therefore natural that the World Health Organization (in the Global Action Plan on Antimicrobial Resistance) recommends that educational efforts should target undergraduates, not just postgraduate education, stating that “making antimicrobial resistance a core component of professional education, training, certification, continuing education and development in the health and veterinary sectors and agricultural practice will help to ensure proper understanding and awareness among professionals.” [14]. Developing an understanding of the current state of undergraduate education in responsible antibiotic use is an essential component of efforts to improve education.
Introduction – Antimicrobial stewardship in Sweden:
A Swedish national action plan was first published in 2000 and has been revised several times, with professional education mentioned in many versions. The 2016 plan [16], following on from the Global Action Plan, stated that the “government expects relevant staff to have knowledge about antibiotic resistance, the spread of infectious diseases and the importance of a high degree of compliance with basic hygiene routines and other infection prevention measures, as well as knowledge about the seriousness and complexity of the issue from a global perspective”; and further that “the government expects antibiotic resistance, infection prevention and control/hygiene to be included in relevant education and training programmes.”
Discussion section:
It is thus clear that there has not been any large-scale improvement in preparedness, despite the growing awareness of antimicrobial resistance and the development and implementation of both the Global Action Plan [14] and National Action Plan [16]. Although both of these plans stressed the importance of undergraduate professional education, to our knowledge, there have been no systematic or top-down efforts to improve education in responsible antibiotic use in medical education since 2015. One potential explanation for this lack of effort is that data on antibiotic prescribing suggest that doctors in Sweden already have very low levels of antibiotic misuse and overuse, in international terms, and current approaches in undergraduate education may have been considered already sufficient. One possibility is that more widespread awareness of the perceived threat of AMR, in part in response to global and national action plans, can have negatively influenced students’ perceptions of their own preparedness to prescribe antibiotics responsibly.
- Add more details about the followed methods for the literature review process and relative references in "Survey Development" in Materials and Methods section
Response: We provided additional information the Materials and Methods section concerning the main 2015 survey, and refer interested readers to the original paper in which the development of the 2015 survey is presented in greater detail. We also add further information concerning the development of the additional questions for the 2021 survey. Finally, we include the 2021 version of the survey as an appendix.
The survey instrument was developed in 2015 by a group of six international experts on antimicrobial stewardship through an informal consensus process, based both on a previous study of undergraduate curricula coverage of principles of prudent antibiotic use in European medical schools [13] and a review of relevant previous studies among medical students. Its complete design, development and piloting (with students from two countries) have been extensively described elsewhere [4]. The 47-item study instrument included questions on demographics, self-reported preparedness in 27 curricular topics on prudent antibiotic use (based on 7-point Likert-type scale principle), opinions of the usefulness of a selection of commonly used teaching methods in undergraduate medical education, and perceived need for further education on this topic. The survey instrument was revised before use in 2021 through the development and addition of nine questions concerning the impact of COVID-19 pandemic on education in responsible antibiotic use, included at the end of the 2015 version of the questionnaire. These questions were de-veloped by the first and last author through a literature review in 2021 to identify other studies that had investigated impacts of Covid-19 on medical student teaching and learning, and the questions were constructed to follow a similar format to the other questions in the original questionnaire. The additional questions were tested for face validity but were not otherwise subjected to pilot-testing, given the low complexity level of the questions.
Reviewer 2 Report
Comments and Suggestions for Authors
The manuscript addresses a relevant and timely topic, as antimicrobial resistance is a global public health threat that requires adequate education and training of future prescribers. Overall, the manuscript is well-written, clear, and informative. It presents the results of a cross-sectional survey of final year medical students in Sweden on their preparedness to prescribe antibiotics responsibly, and compares them with the results of a similar survey conducted in 2015.
Some suggestions for improvement are:
Introduction:
1. The introduction provides a good overview of the topic and the rationale for the study, but it could be improved by including more specific research questions or hypotheses, and by explaining how the study contributes to the existing literature on prudent antibiotic use among medical students.
2. Please provide more details on how Swedish national action plan relates to undergraduate education. Please explain the rationale for repeating the survey in 2021 and the specific research questions or hypotheses.
Methods:
1. Please describe the sampling strategy and the recruitment process of the participants. Explain how the survey instrument was revised and validated for 2021.
2. The authors should also provide the full questionnaire as an appendix or a supplementary file.
Results
1. Please provide more descriptive statistics on the demographic characteristics of the participants and the teaching methods used.
Author Response
Dear reviewer,
Thank you for your kind and helpful comments, we have now reviewed our paper taking into account your recommendations.
Reviewer two:
The manuscript addresses a relevant and timely topic, as antimicrobial resistance is a global public health threat that requires adequate education and training of future prescribers. Overall, the manuscript is well-written, clear, and informative. It presents the results of a cross-sectional survey of final year medical students in Sweden on their preparedness to prescribe antibiotics responsibly, and compares them with the results of a similar survey conducted in 2015.
Some suggestions for improvement are:
Introduction:
- The introduction provides a good overview of the topic and the rationale for the study, but it could be improved by including more specific research questions or hypotheses, and by explaining how the study contributes to the existing literature on prudent antibiotic use among medical students.
Response: We have revised the section “Aim” to more clearly specify our research questions and their connection with (key limitations in) the existing literature.
Previous cross-sectional surveys suggest that medical students in Europe, and beyond, do not feel sufficiently well prepared to prescribe antibiotics responsibly. A key limitation of previous studies is that they have typically captured data at a single timepoint [4, 17-19], meaning that insights are lacking on the longer-term stability of outcomes from such studies; evidence of stable preparedness levels, in the absence of widespread changes in teaching, would further validate the ability to draw conclusions from studies conducted at a single timepoint. In addition, many previous investigations have only assessed a narrow range of curriculum topics relevant to responsible antibiotic use. In 2015, we conducted a large-scale survey study of final year medical students at all European medical schools [4]. Compared with other respondents, Swedish medical students reported very high preparedness levels across a comprehensive range of curriculum topics in responsible antibiotic prescribing, and low levels of perceived need for further education. The main aim of the present study was to build on the previous literature by assessing the stability of these previous findings, both for overall preparedness levels and preparedness in individual topics, through repeating the same 2015 survey with final year medical students in Sweden in 2021. In addition, we aimed to capture potential impacts of the COVID-19 pandemic on education and preparedness within responsible antibiotic use. Our primary research questions were: Are the previously observed high overall preparedness levels among final year Swedish students stable, i.e. similar in 2021 to 2015? Have there been widespread improvements or declines in preparedness levels for individual curriculum topics? Do final year students believe that the Covid-19 pandemic has affected the form, quality and duration of education on antibiotic prescribing at medical school?
- Please provide more details on how Swedish national action plan relates to undergraduate education. Please explain the rationale for repeating the survey in 2021 and the specific research questions or hypotheses.
Response: We have explained the rationale for repeating the survey in 2021 and specific research questions in our response to the preceding question. We have included additional information on Global and National action plans in the introduction session, and reflect further on the relevance to undergraduate education in the discussion section.
Introduction:
It is therefore natural that the World Health Organization (in the Global Action Plan on Antimicrobial Resistance) recommends that educational efforts should target undergraduates, not just postgraduate education, stating that “making antimicrobial resistance a core component of professional education, training, certification, continuing education and development in the health and veterinary sectors and agricultural practice will help to ensure proper understanding and awareness among professionals.” [14]. Developing an understanding of the current state of undergraduate education in responsible antibiotic use is an essential component of efforts to improve education.
Introduction – Antimicrobial stewardship in Sweden:
A Swedish national action plan was first published in 2000 and has been revised several times, with professional education mentioned in many versions. The 2016 plan [16], following on from the Global Action Plan, stated that the “government expects relevant staff to have knowledge about antibiotic resistance, the spread of infectious diseases and the importance of a high degree of compliance with basic hygiene routines and other infection prevention measures, as well as knowledge about the seriousness and complexity of the issue from a global perspective”; and further that “the government expects antibiotic resistance, infection prevention and control/hygiene to be included in relevant education and training programmes.”
Discussion section:
It is thus clear that there has not been any large-scale improvement in preparedness, despite the growing awareness of antimicrobial resistance and the development and implementation of both the Global Action Plan [14] and National Action Plan [16]. Although both of these plans stressed the importance of undergraduate professional education, to our knowledge, there have been no systematic or top-down efforts to improve education in responsible antibiotic use in medical education since 2015. One potential explanation for this lack of effort is that data on antibiotic prescribing suggest that doctors in Sweden already have very low levels of antibiotic misuse and overuse, in international terms, and current approaches in undergraduate education may have been considered already sufficient. One possibility is that more widespread awareness of the perceived threat of AMR, in part in response to global and national action plans, can have negatively influenced students’ perceptions of their own preparedness to prescribe antibiotics responsibly.
Methods:
- Please describe the sampling strategy and the recruitment process of the participants. Explain how the survey instrument was revised and validated for 2021.
Response: We have made some additions to the section “Survey distribution and participants” to explain the recruitment process of participants more clearly. All students in their final term, or second from final term, in autumn 2021 were eligible to participate – and all eligible students were invited by email by a local medical school coordinator. This followed the same protocol as for the 2015 survey:
We followed the same steps for recruitment of participants and survey distribution for the 2021 survey as in the study protocol from 2015 [4]: the “program committee” at each medical school was contacted during early autumn 2021 to identify a local coordinator with responsibility for sending invitations to all eligible medical students, i.e. students in their final term or second last term of medical school at a Swedish university during autumn 2021. Template invitations were drafted by the central study coordinator and forwarded on to medical school coordinators for distribution via email to students. The medical school coordinators were also asked to provide the exact number of eligible students per term, in order to be able to calculate response rates. The initial invitation with survey link was sent by each medical school coordinator by email to all eligible medical students, and this was followed by two email reminders after 2-3 and 4-6 weeks. The self-administered survey was accessible via SurveyMonkey® during spring 2015 (for approximately five months) and autumn 2021 (for approximately three months). Informed consent was obtained from all subjects involved in the study. Participation was voluntary and anonymous.
We also add further information concerning the development of the additional questions for the 2021 survey in the section ”Survey development”:
The survey instrument was developed in 2015 by a group of six international experts on antimicrobial stewardship through an informal consensus process, based both on a previous study of undergraduate curricula coverage of principles of prudent antibiotic use in European medical schools [13] and a review of relevant previous studies among medical students. Its complete design, development and piloting (with students from two countries) have been extensively described elsewhere [4]. The 47-item study instrument included questions on demographics, self-reported preparedness in 27 curricular topics on prudent antibiotic use (based on 7-point Likert-type scale principle), opinions of the usefulness of a selection of commonly used teaching methods in undergraduate medical education, and perceived need for further education on this topic. The survey instrument was revised before use in 2021 through the development and addition of nine questions concerning the impact of COVID-19 pandemic on education in responsible antibiotic use, included at the end of the 2015 version of the questionnaire. These questions were developed by the first and last author through a literature review in 2021 to identify other studies that had investigated impacts of Covid-19 on medical student teaching and learning, and the questions were constructed to follow a similar format to the other questions in the original questionnaire. The additional questions were tested for face validity but were not otherwise subjected to pilot-testing, given the low complexity level of the questions.
- The authors should also provide the full questionnaire as an appendix or a supplementary file.
Response: Thank for your this suggestion, we completely agree. We include the 2021 version of the survey as an appendix.
Results
- Please provide more descriptive statistics on the demographic characteristics of the participants and the teaching methods used.
Response: We are unable to provide more descriptive demographic characteristics of the participants since we did not collection more detailed information, so as to avoid collecting personally identifiable information (a risk that could occur with a very low response rate at an individual medical school). For teaching methods used, we have added columns to table 5 to indicate which methods were not considered either neutral or not useful. We have chosen not to analyse relationships between demographics and perceived usefulness of teaching methods as this was not considered to be a main aim of our research.
Reviewer 3 Report
Comments and Suggestions for Authors
Evaluation of antibiotics prescribing preparedness at the end of Medical School is an important aspect of the medical curriculum. Here, is proposed a substitute with a self-evaluation administered through a detailed questionnaire of 27 topics. However, there is a great difference between objective evaluation by teaching staff and self-evaluation of medical students. The limits of the second option could be addressed and explained.
TAB 1 and 2 are long tables but their differences are not very clear: could they be merged in only one table?
TAB 6 presents different teaching methods divided between useful/very useful and not used: why the sum of each percentage is never 100%? For example: lectures (with > 15 people) in 2015: “useful/very useful” 88.1% and “not used” 0.4%. The sum 88.1 + 0.4 is = 88,5%; the remaining11.5% did not answer??
Author Response
Dear reviewer,
Thank you for your kind and helpful comments, we have now reviewed our paper taking into account your recommendations.
Reviewer 3
Evaluation of antibiotics prescribing preparedness at the end of Medical School is an important aspect of the medical curriculum. Here, is proposed a substitute with a self-evaluation administered through a detailed questionnaire of 27 topics. However, there is a great difference between objective evaluation by teaching staff and self-evaluation of medical students. The limits of the second option could be addressed and explained.
Response: We do agree, and have noted this limitation in conjunction with our previous study. Since our 2021 survey was a follow-up (with an emphasis on comparison between-years) we have emphasized this aspect less in the current manuscript, but agree that it should be noted. We have thus added a few sentences at the end of the section “Methodological considerations”:
We have previously noted [4, 18] that an important consideration when interpreting our findings is that the questionnaire was designed to elicit students’ self-reported preparedness levels within each curriculum topic, as opposed to objectively assess individual student’s actual preparedness level. Additional studies are required to understand how these aspects are related, and whether the strength of the relationship varies between different topics. Nonetheless, we contend that gaining student perspectives on teaching is a vital component of quality improvement efforts, and fully in keeping with recommended student-centred practices in higher education [Biggs and Tang BOOK].
TAB 1 and 2 are long tables but their differences are not very clear: could they be merged in only one table?
Response: We agree the tables are long, and it is hard to improve their presentation without removing results. We have decided to include the range in differences between 2015 and 2021 in Table 1, and to move Table 2 to the appendix, for readers that would like to find out about the finer resolution results on variations at individual medical schools. We have also coloured alternating rows in Table 1 in light grey to improve readability.
TAB 6 presents different teaching methods divided between useful/very useful and not used: why the sum of each percentage is never 100%? For example: lectures (with > 15 people) in 2015: “useful/very useful” 88.1% and “not used” 0.4%. The sum 88.1 + 0.4 is = 88,5%; the remaining11.5% did not answer??
Response: We had previously decided to show only the results for “useful/very useful” and “not used”, to simplify presentation in the table, and because we felt these were the most interesting results. We have now updated this table for clarity, incorporating all response options. We have also added a footnote so that it can be understood that the denominator for “usefulness” results does not include students who reported that the method was not used – i.e. the results for “usefulness” add up to 100%.